# Moxibustion for declined cardiorespiratory fitness of apparently healthy older adults: A study protocol for a randomized controlled trial

**Zheng Sun** [1‡], **Zhihong Xu** [2‡], **Kuang Yu** [1], **Haitian Sun** [1], **Yiren Lin** [2], **Zilong Zhu** [1], **Yimin Zhu** [3], **Jianbin Zhang** [1] *

1 Department of Acupuncture and Moxibustion, The Second Affiliated Hospital of Nanjing University of Chinese Medicine, Nanjing, China, 2 Chinese Medicine Rehabilitation Center, The Second Affiliated Hospital of Nanjing University of Chinese Medicine, Nanjing, China, 3 Department of Pulmonology, The Second Affiliated Hospital of Nanjing University of Chinese Medicine, Nanjing, China

‡ ZS and ZX contributed equally to this work as co-first authors.
* zhangjianbin@njucm.edu.cn

**Data Availability Statement:** No datasets were generated or analysed during the current study. All relevant data from this study will be made available upon study completion.

## Abstract

### Background

Aging and age-related declines lead to varying degrees of decreased cardiorespiratory fitness (CRF) in apparently healthy older adults. Exercise training, the primary approach for enhancing CRF, encounters several constraints when used with elderly individuals. Existing evidence implies that moxibustion might enhance the CRF of older adults. However, clinical research in this area still needs to be improved.

### Methods

This study will employ a randomized, assessor-blinded, controlled trial design involving 126 eligible participants. These participants will be stratified and randomly assigned to one moxibustion group, one sham moxibustion group, and one blank control group. Acupoints of bilateral *Zusanli* (ST36), *Shenque* (CV8), and *Guanyuan* (CV4) are selected for both real and sham moxibustion groups. The treatment will last 60 min per session, 5 sessions a week for 12 weeks. The blank control group will not receive any intervention for CRF improvement. Primary outcomes will be the mean change in peak oxygen uptake ($VO_{2peak}$), anaerobic threshold (AT), and serum central carbon metabolites (CCB) from the baseline to observation points. Secondary outcome measures involve the six-minute walk distance (6MWD), the Short Form 36 Health Survey (SF-36), and the Qi and Blood Status Questionnaire (QBSQ). Outcome assessments will be conducted at weeks 4, 8, 12, and 24 as part of the follow-up. Adverse events will be assessed at each visit.

### Discussion

This trial can potentially ascertain moxibustion's effectiveness and safety in enhancing CRF among apparently healthy older adults.

**Funding:** This study was funded by a government grant "Jiangsu Traditional Chinese Medicine Science and Technology Development Project" from the Jiangsu Commission of Health (grant number ZT202208). The funders had no role in study design, data collection and analysis, publication decisions, or manuscript preparation.

**Competing interests:** The authors have declared that no competing interests exist.

**Abbreviations:** CRF, Cardiorespiratory fitness; TCM, Traditional Chinese Medicine; $VO_{2peak}$, Peak oxygen uptake; AT, Anaerobic threshold; CCB, Central carbon metabolism; QBSQ, Qi and Blood Status Questionnaire; 6MWD (T), Six-minute walk distance (test); SF-36, 36-Item Short Form Health Survey; CPET, Cardiopulmonary Exercise Test; ACSM, The American College of Sports Medicine; ITT, Intension to treat; LOCF, Last Observation Carried Forward; CRFs, Paper case report forms.

## Trail registration

ChiCTR, ChiCTR2300070303. Registered on April 08, 2023.

## Introduction

Cardiorespiratory fitness (CRF) is considered a fundamental aspect of health-related fitness and has been acknowledged by the American Heart Association as a novel vital sign [1]. Long-term clinical studies with follow-up have consistently demonstrated an inverse relationship between cardiorespiratory fitness and all-cause mortality [2–4].

Studies have cleared that one of the most significant physiological changes resulting from aging is the decrease in maximum aerobic capacity, indicated by the reduction in peak oxygen consumption, reflecting the decline in CRF [5, 6]. This decline exhibits a non-linear exacerbation pattern with advancing age, with the rate of decline surpassing 20% per decade after reaching the age of 70 [7]. Potential reasons may involve the natural decline of organ function due to aging [8, 9] and a sedentary lifestyle.

Exercise training is currently the primary way to enhance CRF, but it requires specific prerequisites. One review [10] detailed specific endurance training parameters to be effective for enhancing CRF in sedentary older adults, including intensity (66%-77% of Heart Rate Reserve), frequency (3–4 days per week), duration (40–50 minutes per session), and program length (30–40 weeks). Nonetheless, a survey conducted in the United States [11] revealed that only 22% of adults aged 65 and older engaged in regular physical activity, and this percentage declined with age. Italy exhibited even lower engagement rates [12]. Consequently, advocating for exercise therapy among older adults may present challenges. Additionally, older adults represent a distinctive demographic experiencing age-related declines in physical function. Hence, multiple limitations arise in formulating exercise prescriptions, involving pre-exercise assessments, regulating intensity, and selecting proper exercise types. Failure to devise an appropriate exercise prescription may heighten the risk of injury [13].

Moxibustion is a traditional Chinese external therapy that involves treating diseases by applying heat generated by burning leaves of Artemisia argyi to specific acupoints on the body's surface. Our previous research [14, 15] showed that moxibustion can elevate skin temperatures to 45˚C, and this kind of thermal stimulation has been shown to induce local and distal vasodilation and increase blood circulation, which is necessary for improving CRF. In addition, clinical studies have demonstrated that moxibustion can enhance pulmonary function in patients with COPD [16, 17] and cardiac function in individuals with heart failure [18] and coronary heart disease [19]. However, whether moxibustion can improve CRF in apparently healthy older adults is unknown. Therefore, this study protocol was developed as a randomized controlled trial to determine moxibustion's clinical effectiveness and safety in enhancing CRF in apparently healthy older adults.

## Methods/Design

### Study design

This is a randomized, controlled, assessor-blinded, parallel-group trial. The enrollment, interventions, and assessment schedule can be seen in **Fig 1**. After screening, eligible participants who provide signed informed consent will be randomly allocated to one moxibustion group, one sham moxibustion group, and one blank control group in a 1:1:1 ratio. Baseline

| Study Period | | | | | | | |
|---|---|---|---|---|---|---|---|
| | **Enrolment** | **Allocation** | **Treatment period** | | | | **Follow-up** |
| **Timepoint** | $-Wk_1$ | $Wk_0$ | $Wk_1$ | $Wk_4$ | $Wk_8$ | $Wk_{12}$ | $Wk_{24}$ |
| **Entrolment** | | | | | | | |
| Eligible screen | X | | | | | | |
| Infromed consent | X | | | | | | |
| Basline assessement | X | | | | | | |
| Allocation | | X | | | | | |
| **Intervention** | | | | | | | |
| Moxibustion | | | ←————————————→ | | | | |
| Sham moxibustion | | | ←————————————→ | | | | |
| Control | | | ←————————————→ | | | | |
| **Assessments** | | | | | | | |
| $VO_{2peak}$ | X | | | X | X | X | X |
| AT | X | | | X | X | X | X |
| Serum CCM | X | | | X | X | X | X |
| 6MWD | X | | | X | X | X | X |
| SF-36 | X | | | X | X | X | X |
| QBSQ | X | | | X | X | X | X |
| **Adverse events** | | | ←————————————→ | | | | |

**Fig 1. Schedule for treatment and outcome assessments.** $VO_{2peak}$, Peak oxygen uptake; AT, Anaerobic threshold; 6MWD, Six-minute walk distance; SF-36, 36-Item Short Form Health Survey; QBSQ, TCM Qi and Blood Status Questionnaire.

measurements will occur before the initial intervention commences. Outcome measures and adverse events will be assessed at weeks 4, 8, and 12 after the initiation of moxibustion intervention, with a follow-up visit scheduled at week 24. The flow chart of the trial is shown in **Fig 2**. The SPIRIT checklist is provided in S1 File.

In this trial, both the real and sham moxibustion groups will target the same acupoints. However, the sham group will utilize a specifically designed device resembling the authentic moxibustion device to isolate heat and radiation produced during moxa burning. The blank control group will not receive moxibustion treatment or any other intervention that may improve CRF. The primary outcome measures are peak oxygen uptake ($VO_{2peak}$), anaerobic

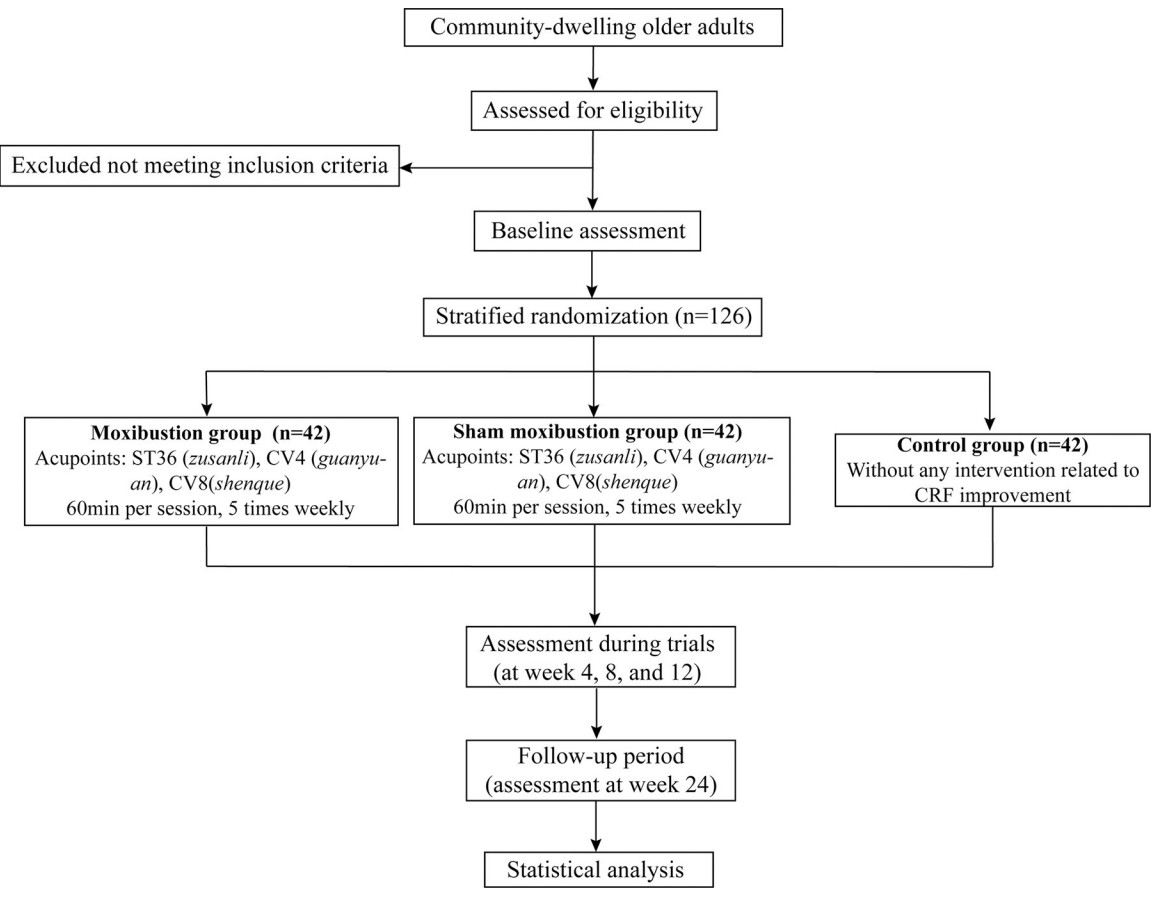

**Fig 2. Flow chart of study design.**

threshold (AT), and serum central carbon metabolites (CCB), and the secondary outcome measures are six-minute walk distance (6MWD), the Short Form 36 Health Survey (SF-36), and the Qi and Blood Status Questionnaire (QBSQ).

## Ethics statement and registry

The study protocol has been approved by the Ethics Committee of the Second Affiliated Hospital of Nanjing University of Traditional Chinese Medicine (Approval no. 2023SEZ-007-01) (S2 File) and registered in ChiCTR (ChiCTR2300070303) on April 08, 2023. Written informed consent will be obtained from each participant at the beginning of the trial.

## Participants

**Setting.** All participants with declined CRF recruited through advertisements and the neighboring community of the Second Affiliated Hospital of Nanjing University of Traditional Chinese Medicine who comply with the following criteria:

**Inclusion criteria.** Participants who meet all the following criteria are eligible for the study:

- Men or women aged between 60 and 80;

- Apparently healthy without underlying chronic diseases, such as chronic obstructive pulmonary disease, hypertension, diabetes, musculoskeletal pain, etc.;

- Decreased CRF ($VO_{2peak}$ < 20ml·kg-1·min-1 or 6MWD ≤ 450 meters).

  **Exclusion criteria.** Participants will be excluded if they have the following conditions:

- With previous moxibustion history or experience;

- Currently taking supplements that could influence the study results, such as coenzyme Q10;

- Respiratory symptoms within one month;

- Other conditions that may reduce the possibility of enrollment or complicate enrollment, such as frequent changes in the living environment, are likely to cause a loss of follow-up.

## Sample size calculation

Sample size determination was based on a one-way analysis of variance (ANOVA) comparing the $VO_{2peak}$ value of the three different groups. A Total sample of 107 participants is required to detect a significant difference with an effect size of 0.35 and 90% power at a 5% level of significance. Considering an estimated dropout rate of 15%, approximately 126 participants in total are needed, with 42 for each group. Sample calculation was conducted using G-Power software version 3.1.9.4 (Franz Faul, Uni Kiel, Germany). To recruit this number of participants, we anticipate a study period of 9 months based on our pilot study.

## Randomization and allocation concealment

Random numbers will be generated using SPSS (Version 27.0), sealed in sequentially numbered opaque envelopes, and delivered to practitioners. After obtaining informed consent from eligible participants, the practitioners will unseal the envelopes and assign individuals to either the moxibustion group, the sham moxibustion group, or the blank control group based on the contents.

This study employs a stratified random grouping approach to ensure comparability of baseline conditions among participants in each group. Following the Weber and Ventilatory Classification Systems [20], the participants' CRF will be categorized into three grades: grade II (16ml·kg-1·min-1≤$VO_{2peak}$<20ml·kg-1·min-1), grade III (10ml·kg-1·min-1≤$VO_{2peak}$<16ml·kg-1·min-1), and grade IV ($VO_{2peak}$<10ml·kg-1·min-1). The participants in each grade are divided into three groups using a simple randomization method. Then, they were assigned to either the moxibustion group, the shame moxibustion group, or the blank control group. This allocation approach ensures an even distribution of participants with varying degrees of declined CRF across three groups. The stratified random grouping process is visually represented in **Fig 3**.

## Blinding

The nature of the moxibustion intervention precludes the possibility of achieving blinding between intervention and blank control groups. However, blinding will be implemented for participants assigned to the real or sham moxibustion groups. Specifically, participants in these two groups will be allocated to individual isolation treatment rooms, refraining from communicating with one another. Besides, the outcome evaluators and those responsible for data collection and statistical analysis will be kept unaware of the grouping information and specific allocation details about the entire trial. Unblinding of the study will happen if a serious adverse event occurs or upon the completion of research, data analysis, and interpretation.

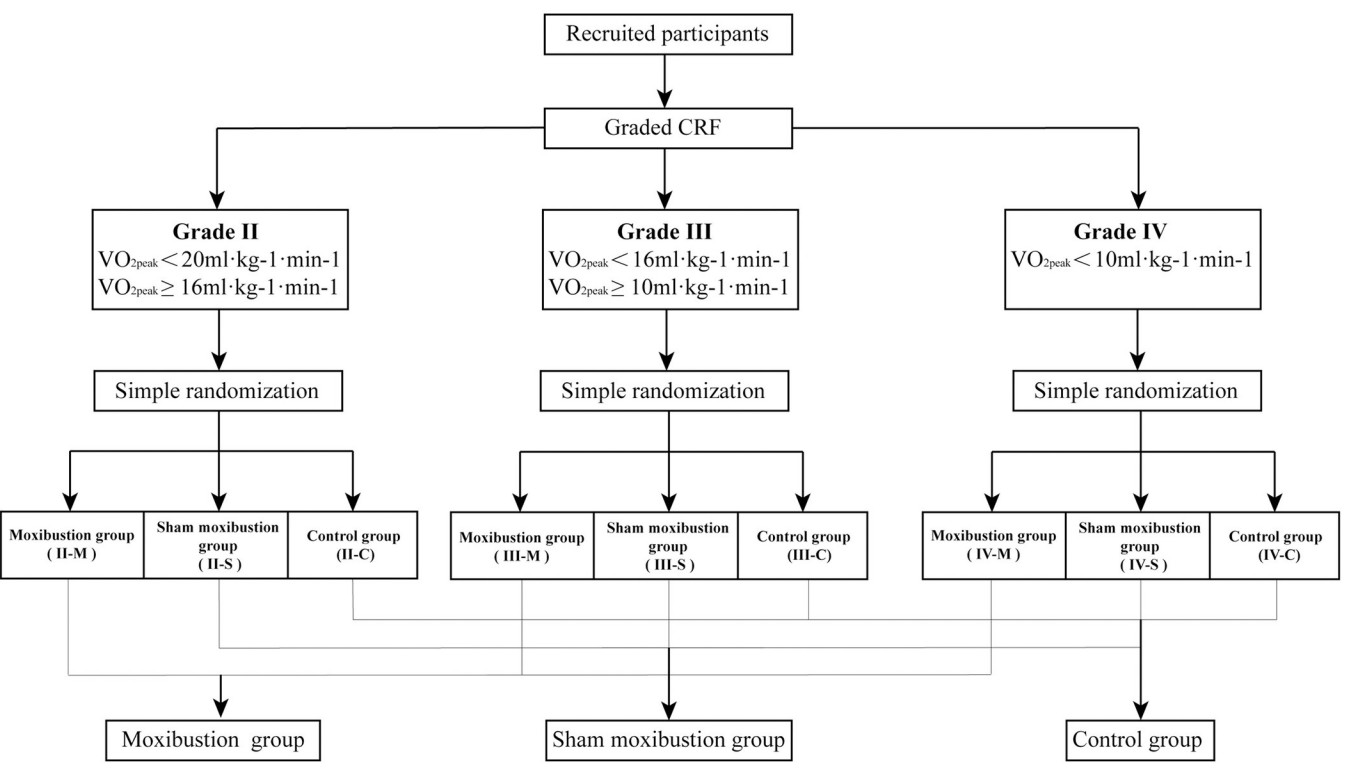

**Fig 3. The stratified randomization process.**

## Intervention

This trial comprises three arms. In addition to the intervention we designed and matched, participants in each group will not receive any other intervention. They are requested to maintain their original lifestyle, including exercise routines, dietary patterns, work schedules, and rest habits.

**Moxibustion group.** Based on our previous research [21], this trial will employ the acupoints, including bilateral *Zusanli* (ST36), *Shenque* (CV8), and *Guanyuan* (CV4). Acupoint positioning refers to the guidelines outlined in the 2006 National Standard of the People's Republic of China, "Acupoint Names and Locations" (GB/T12346-2006): The ST36 is located at 3 *cun* (about 10 cm) below the patella, outside of the anterior crest of the tibia. CV8 is located at the navel, and CV4 is 3 *cun* directly below the CV8 (**Fig 4**).

Moxibustion Operation:

- The patient lies supine, exposing the skin at the selected acupoints, hands placed on either side of the body.

- The practitioners identify the acupoints and, after marking their locations on the local skin with a marker, secure the moxibustion device base onto the selected acupoints using adhesive tape.

- Ignite the moxa stick on the moxibustion cylinder.

- Position the moxibustion device cylinder so that its top is level with the bottom of the device base.

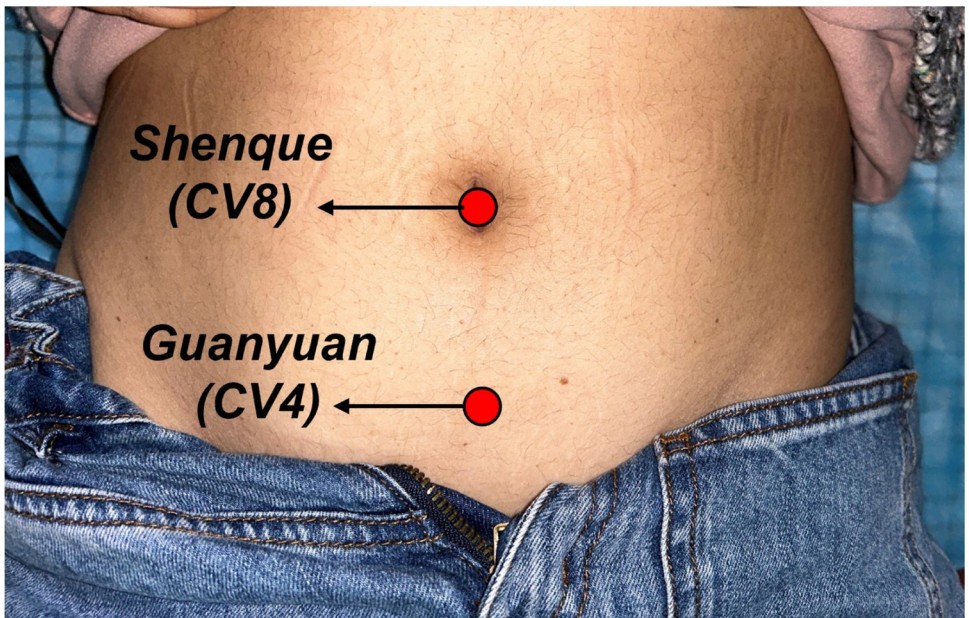

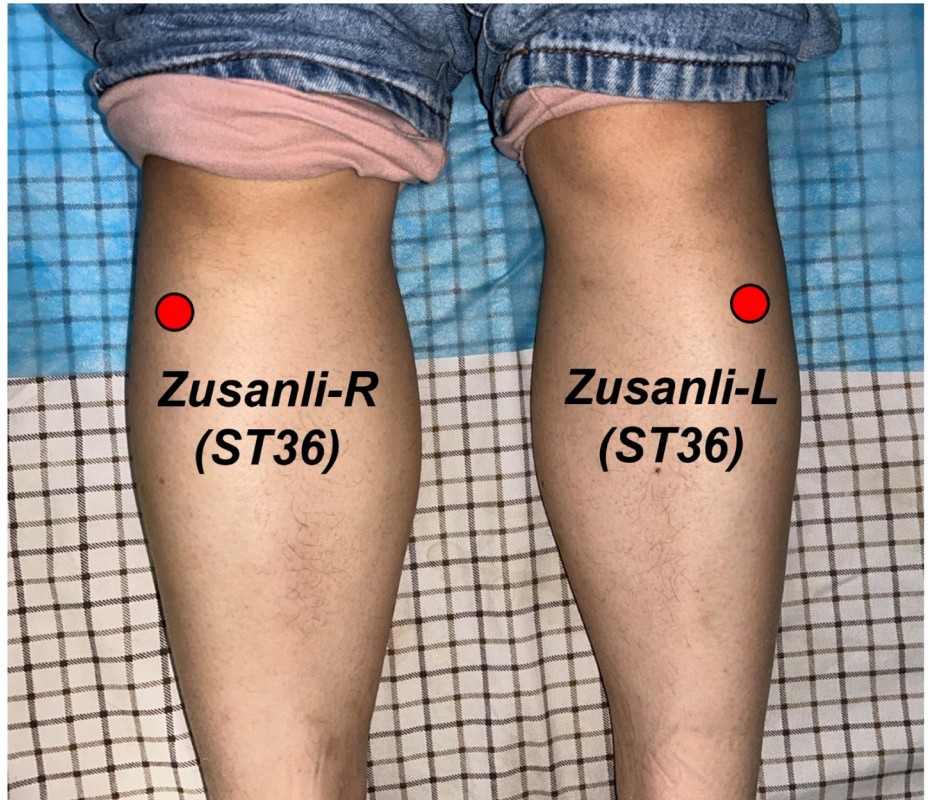

**Fig 4. Acupoints were applied in the trial.** R, right side; L, left side. (Fig 4 was taken by the author. The individual in the figure has granted permission in paper publication under CC BY license).

- Rotate the moxibustion cylinder to adjust the air outlet size, maintaining a distance of approximately 3cm from the skin; make participants feel comfortably warm without burning pain.

- When the warm feeling diminishes, indicating the moxa stick is burnt out, it can be replaced.

- After 60 minutes of moxibustion, close the air outlet, remove the moxibustion device, and place the extinguished moxa stick in a container with water.

- Check the moxibustion device and store it back in the original packaging for future use.

- Each participant undergoes moxibustion once a day (8:00–12:00 AM or 2:00–5:00 PM), 60 minutes per acupoint, five times a week (with weekends off), 12 weeks (60 sessions) in total.

**Sham moxibustion group.** We designed a sham moxibustion group as a placebo control. Participants in this group will receive sham moxibustion treatment at the same acupoints as the moxibustion group. We will use a sham moxibustion device, which resembles the real one, but the base has a metal membrane to isolate heat and smoke generated by the burning moxa and prevent it from radiating to the skin (**Fig 5**).

The operation of the sham moxibustion group will be basically the same as those of the moxibustion group, except for the following steps, which are slightly different:

- Use the base of the shame moxibustion device (with a metal membrane).

- Keep the distance between the moxa stick burning end and the skin at about 5cm instead of 3cm.

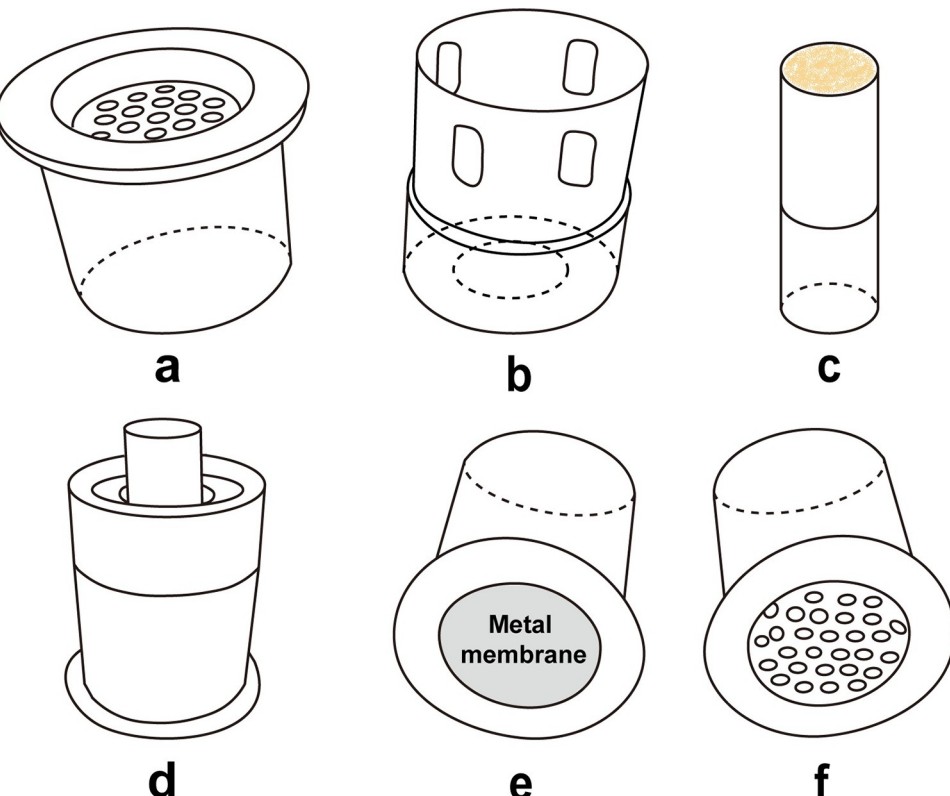

**Fig 5. Ignition-type moxibustion device. a**, the base of the device; **b**, the body of the device; **c**, moxa stick; **d**, the assembly of a, b, and c; **e**, bottom view of sham moxibustion device base; **f**, bottom view of real moxibustion device view. (Fig 5 was created by the author using Adobe Illustrator 2023 based on the picture of the moxibustion device).

- Replace the moxa stick after 30 minutes instead of waiting for the warmth sensation to decrease.

**Blank control group.**   The blank control group will remain untreated and be instructed to refrain from any treatments that may impact CRF until the trial is completed. Outcome measures of the blank control group will be conducted following the designed time points, consistent with the moxibustion and sham moxibustion groups.

## Dropout and withdrawal

Participants meeting any of the following criteria will be considered as dropouts or withdraw from the trial, and specific reasons will be thoroughly documented:

Dropout:

- Participants are unwilling to continue the clinical trial due to personal reasons;

- Unforeseen circumstances hindering subject participation, such as severe car accidents, sudden grave illness, or fatality.

Withdrawal:

- Serious adverse events related to moxibustion, such as skin burns leading to infection;

- Inadequate treatment compliance, defined by a lower than anticipated actual-to-planned treatment ratio (70%);

- Significant alterations in lifestyle habits, including dietary changes, a transition from sedentary to active routines, or vice versa;

- Weight changes exceeding 5 kilograms.

## Outcome measures

The following outcome measures will be assessed by assessors blinded to group allocations.

**Primary outcome measures.**   The primary assessment of decreased CRF effectiveness will involve calculating the mean changes in $VO_{2peak}$, AT, and serum CCB at 4, 8, and 12 weeks from baseline.

Peak or Maximal oxygen uptake ($VO_{2peak}$ or $VO_{2max}$) is an established measure indicating the level of CRF. It serves as a diagnostic tool for evaluating reduced CRF. However, research indicates that $VO_{2max}$ measurement is challenging for most older individuals due to their limited ability to endure extremely intense exercise loads [22]. Consequently, $VO_{2peak}$ is often used as an estimate for $VO_{2max}$. For practical purposes, $VO_{2max}$ and $VO_{2peak}$ are used interchangeably [23]. AT, also known as the lactate threshold, is considered an estimator of the $VO_2$ at which the rate of change in arterial lactate rapidly increases during exercise, playing a critical role in determining one's endurance capacity. Higher AT indicates an individual's ability to exercise at a higher intensity before fatigue. In this study, both $VO_{2peak}$ and AT will be directly measured through Cardiopulmonary Exercise Testing (CPET) using Smax58ce-sp (Hanya Health Technology Co., Ltd. Nanjing, China). The CPET will follow an individualized protocol on a cycle ergometer (Cardiowise cycle ergometer, V6001-0004, Germany), adhering to the ATS/ACCP Statement on Cardiopulmonary Exercise Testing [23].

Central carbon metabolism (CCB), known as energy metabolism, primarily involves biological processes of the tricarboxylic acid cycle (TAC), glycolysis, and pentose phosphate

pathway at the cellular level. These pathways consume oxygen and generate the energy needed for body activity. During physical exercise, stress responses in the respiratory and cardiovascular systems are triggered and correlated with increased cellular CCB levels. This study aims to examine 34 serum CCB-related biomarkers, such as ATP, $NAD^+$, Pyruvic Acid, and Acetyl-CoA. The analysis will use an ultra-high performance liquid chromatography coupled to tandem mass spectrometry (UHPLC-MS/MS) system (ExionLC™ AD UHPLC-QTRAP® 6500+, AB SCIEX Corp., Boston, MA, USA). Venous blood samples will be collected from participants and centrifuged at 3000 rpm and 4˚C for 10 minutes. 0.2 ml of serum will be extracted into a 1.5 ml centrifuge tube. After labeling, the samples will be frozen in liquid nitrogen for 15 minutes and stored in an ultra-low temperature refrigerator (-80˚C). Serum samples will be supervised by the Central Laboratory of the Second Affiliated Hospital of Nanjing University of Chinese Medicine for subsequent testing.

**Secondary outcome measures.** Secondary outcomes involve the average change in 6MWD and the overall scale value of the SF-36 and QBSQ at 4, 8, and 12 weeks compared to the baseline.

The 6-minute walk test (6MWT) is a commonly employed sub-maximal exercise test that evaluates an individual's CRF by measuring the distance covered over 6 minutes. The 6MWT is an optimal supplemental method for assessing CRF in older adults, as the 6MWD exhibits a strong correlation with $VO_{2peak}$ measured through CPET [24, 25]. The conduction of 6MWT in this study will follow the ATS Statement: guidelines for the six-minute walk test [26].

The SF-36 assesses the general health status and quality of life in patients and non-patients, comprising 36 questions across 8 categories, where higher scores signify better functional levels. The domains of Physical Functioning, Role-Physical, General Health, Vitality, and Social Functioning in SF-36 delineate the multifaceted impact of CRF on quality of life [27–29].

The QBSQ employs standardized queries from the Constitution in Chinese Medicine Questionnaire, covering four components: Qi-deficiency, Qi-stagnation, Blood-deficiency, and Blood-stasis, comprising 29 questions, with 8 focusing on Qi-deficiency and 7 on each of the other three components. The scoring system assigns numerical values (1 to 5) to five response options ("none, little, sometimes, often, and always") for each question. Our preliminary study observed a negative linear correlation between QBSQ scores and 6MWD, suggesting that higher QBSQ scores may indicate lower CRF.

## Statistical analysis

An independent statistician blinded to group allocation will conduct the statistical analysis. Analyses for efficacy and safety will follow the intention-to-treat (ITT) principle. Missing values will be addressed using the last observation carried forward (LOCF) method. Each group's demographic characteristics and baseline variable measurements will be summarized. The Kolmogorov-Smirnov test will be applied for data normality assessment. Continuous data will be presented as mean ± standard deviation for normally distributed data and median (range) for non-normally distributed data. Paired t-tests or analysis of variance (ANOVA) will be employed to assess changes in primary and secondary outcomes before and after intervention. Pairwise comparison of primary and secondary outcomes between groups will be analyzed using ANOVA or the Kruskal-Wallis test. The Chi-square or Fisher's exact test will be used to analyze categorical data. The correlations between variables will be evaluated using Pearson or Spearman correlation analyses. The Statistical Package for the Social Sciences (SPSS) version 27.0 for Windows will be employed for statistical analysis, with significance set at a two-sided P-value of $< 0.05$. The graph analysis will be performed using GraphPad Prism software (Version 9.4.0).

## Protocol amendments

Important protocol adjustments, such as alterations to eligibility criteria, outcome measures, or analysis, will be communicated to relevant parties, including the Research Ethics Committee, researchers, participants, and the publishing journal.

## Data collection, monitoring, and quality control

Participant characteristics will be gathered during the baseline assessment conducted before randomization. All data will be obtained using pre-designed and printed Paper case report forms (CRFs). One research assistant will scan the original data as image files and share them with another research assistant. Two research assistants fill the data into a Microsoft Excel spreadsheet separately and verify the integrity and accuracy of the data. Any discrepancies will be investigated and resolved by reviewing the original data.

Each participant will be assigned an individual trial number to maintain confidentiality. The paper-form data will be stored in locked filing cabinets at the office of the Acupuncture and Moxibustion Department director. Electronic-form data will be securely stored on a password-protected computer. Data will only be accessible to the research team. The independent ethics committee of the Second Affiliated Hospital of Nanjing University of Chinese Medicine will oversee the data. The monitors assigned to this task will assess the accuracy of the information recorded on the paper CRFs and ensure that the trial procedures follow the established protocol (S1 Protocol).

To ensure the trial's quality, all researchers must complete a training program covering various aspects, including trial planning, participant eligibility criteria, intervention methods for each group, outcome measures, and accurate documentation of paper CRFs. All practitioners hold doctor qualifications in Traditional Chinese Medicine and possess at least three years of clinical experience in moxibustion therapy.

To enhance participant compliance, we adopt the following strategies:

- Timely reminders for visits;

- Waiving all treatment and assessment costs;

- Offering financial compensation to participants completing all planned treatments and assessments.

Comprehensive documentation will detail reasons and outcomes for participants discontinuing trial involvement or failing to adhere to the follow-up process.

## Adverse events

Participants and practitioners will report any unforeseen and unintended reactions resulting from moxibustion treatment, including but not limited to skin burns, blisters, itching, and respiratory symptoms. The symptoms, onset time, duration, and severity of adverse events will be carefully documented in the CRFs. Practitioners will assist in managing adverse events and assess whether treatment suspension is necessary.

## Discussion

This study was designed to evaluate the efficacy and safety of moxibustion treatment on apparently healthy older adults with declined CRF.

In this study, we select apparently healthy older adults as our research objective because our preclinical observation indicated that a notable percentage of apparently healthy community-

dwelling older adults exhibited varying degrees of diminished CRF during their CPET. CRF extensively indicates the function of various organs, such as the lungs, heart, blood vessels, and muscles [30]. Studies have confirmed that aging is an independent risk factor for these organs' detrimental structural and functional alternations [8, 31–34]. Malfunctions in any of these organs can impact oxygen intake, transportation, and utilization, decreasing maximal/peak oxygen uptake ($VO_{2max}$/$VO_{2peak}$) [35]. No studies employ CPET to evaluate CRF among healthy older adults in China. In 2020, Zou et al. [36] conducted a survey that presented the 6MWD of 266 healthy Chinese individuals aged 60 years and above. Nonetheless, the study only provided the mean value of the 6MWD in healthy older adults without reporting the count or percentage of abnormalities.

Lung respiration is interconnected with cellular respiration via the cardiovascular system [37]. Cellular respiration generates the energy needed for physical activity, with the CCB as its primary pathway [38]. Therefore, CCB-related biomarkers partially reflect the level of cellular respiration and potentially have associations with external lung respiration. By incorporating CCB-related biomarkers as an outcome measure, we aim to observe the subtle effects of moxibustion and perform correlation analysis between $VO_{2peak}$, AT, 6MWD, and CCB-related biomarkers. This effort will offer a foundation for further mechanistic research on how moxibustion enhances CRF.

In this study, we aim to mitigate unwanted baseline variability among groups by considering $VO_{2peak}$ as a stratification factor and employing a stratified randomization method. CRF levels vary considerably in healthy older adults. Data from a Norwegian survey [22] showed that $VO_{2peak}$ ($ml \cdot min^{-1} \cdot kg^{-1}$) values ranged from 19.8–37.3 in healthy older women and from 23.5–46.1 in healthy men (5th-95th percentiles). Therefore, simple randomized grouping may result in unwanted variability in baseline data among groups.

The inclusion of a sham moxibustion group as a placebo control is necessary due to the controversy over the placebo effect of TCM complementary therapy. As a result of numerous clinical studies focusing on sham moxibustion design [39–41] and advancements in sham moxibustion devices [42], the research application of sham moxibustion has been continuously improved. It is known that temperature plays a crucial role in the effectiveness of moxibustion, and isolating heat can notably diminish its effectiveness [43]. Therefore, we will use a modified moxibustion cylinder equipped with a metal membrane capable of isolating heat and radiation while maintaining an appearance indistinguishable from the real one [37, 38].

However, there remain limitations to this trial. Firstly, despite our efforts to optimize blinding procedures, the inherent properties of moxibustion pose challenges in achieving complete practitioner blinding. Consequently, there may have been performance and detection biases to a certain degree. Secondly, our study is a single-center trial conducted solely in one hospital in China; thus, the generalizability and applicability of these findings to other patient populations in different regions still need to be determined.

In conclusion, this randomized controlled trial using a stratified randomization approach and sham moxibustion design may provide rigorous evidence for the effectiveness of moxibustion in addressing decreased CRF in apparently healthy older adults.

## Supporting information

**S1 File. SPIRIT checklist.**
(DOCX)

**S2 File. Ethical approval document (English).**
(PDF)

**S1 Protocol. Study protocol (English).**
(PDF)

## Acknowledgments

The authors are grateful to all collaborators and participants of the study.

## Author Contributions

**Conceptualization:** Zheng Sun, Jianbin Zhang.

**Investigation:** Zheng Sun, Zhihong Xu, Kuang Yu, Haitian Sun.

**Methodology:** Zheng Sun, Zhihong Xu, Yiren Lin, Yimin Zhu.

**Project administration:** Zheng Sun, Jianbin Zhang.

**Supervision:** Jianbin Zhang.

**Writing – original draft:** Zheng Sun, Zhihong Xu, Zilong Zhu, Jianbin Zhang.

**Writing – review & editing:** Zheng Sun, Zhihong Xu, Yiren Lin, Yimin Zhu, Jianbin Zhang.

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
