## [Decision Letter · Decision Letter 0]

2 Feb 2024

PONE-D-23-40936Moxibusiton for declined cardiorespiratory fitness of apparently healthy older adults: a study protocol for a randomized controlled trial

PLOS ONE

Dear Dr. Sun,

Thank you for submitting your manuscript to PLOS ONE. After careful consideration, we feel that it has merit but does not fully meet PLOS ONE’s publication criteria as it currently stands. Therefore, we invite you to submit a revised version of the manuscript that addresses the points raised during the review process.

We look forward to receiving your revised manuscript.

Kind regards,

Yoshihiro Fukumoto

Academic Editor

PLOS ONE

4. We note that Figures 4 and 5 in your submission contain copyrighted images. All PLOS content is published under the Creative Commons Attribution License (CC BY 4.0), which means that the manuscript, images, and Supporting Information files will be freely available online, and any third party is permitted to access, download, copy, distribute, and use these materials in any way, even commercially, with proper attribution. For more information, see our copyright guidelines: http://journals.plos.org/plosone/s/licenses-and-copyright.

1. You may seek permission from the original copyright holder of Figures 4 and 5 to publish the content specifically under the CC BY 4.0 license.

Reviewers' comments:

Reviewer's Responses to Questions

**Comments to the Author**

1. Does the manuscript provide a valid rationale for the proposed study, with clearly identified and justified research questions?

Reviewer #1: Partly

Reviewer #2: Yes

2. Is the protocol technically sound and planned in a manner that will lead to a meaningful outcome and allow testing the stated hypotheses?

Reviewer #1: Partly

Reviewer #2: Yes

3. Is the methodology feasible and described in sufficient detail to allow the work to be replicable?

Reviewer #1: Yes

Reviewer #2: Yes

4. Have the authors described where all data underlying the findings will be made available when the study is complete?

Reviewer #1: Yes

Reviewer #2: Yes

5. Is the manuscript presented in an intelligible fashion and written in standard English?

Reviewer #1: Yes

Reviewer #2: Yes

6. Review Comments to the Author

You may also provide optional suggestions and comments to authors that they might find helpful in planning their study.

Reviewer #1: This is a study protocol to evaluate the efficacy of Moxibusiton in elderly. I have a few comments to the authors.

1. Study rationale: What is the rationale to intervene to healthy elderly without hypertension, diabetes, etc? They are asymptomatic and doing well.

2. Inclusion and exclusion criteria should be provided more clearly.

3. Sample size calculation: this trial has 3 arms but sample calculation was done by 2 arms design. Adding additional 35 patients is missing reasonability.

Reviewer #2: This is a design paper for a study examining the improvement of exercise tolerance through moxibustion.

The reviewer has some questions for this study.

1. Aren't people with experience in moxibustion excluded? If subject with experience in moxibustion is assigned to the Sham group, is there a possibility that the subject will realize that the sham procedure is being used?

2. Isn't the effect of moxibustion different depending on the practitioner? Are differences between practitioners taken into consideration?

3. What kind of consideration is given to placing moxibustion at the exact position of the acupuncture points?

4. What are the possible reasons for withdrawal from research? I think there is a possibility that moxibustion cannot be continued due to pain etc.

7. PLOS authors have the option to publish the peer review history of their article (what does this mean?). If published, this will include your full peer review and any attached files.

Reviewer #1: No

Reviewer #2: No

---

## [Author Response · Author response to Decision Letter 0]

28 Feb 2024

Responses to the reviewers

Reviewer #1: 

This is a study protocol to evaluate the efficacy of Moxibustion in elderly. I have a few comments to the authors.

Our response: We sincerely appreciate your review of our article and the valuable suggestions and comments you provided. Your input plays a pivotal role in enhancing the quality of the paper.

1. Study rationale: What is the rationale to intervene to healthy elderly without hypertension, diabetes, etc? They are asymptomatic and doing well.

Our response: Thank you for this thoughtful question. We selected healthy older adults as research subjects for the following reasons: 1) Even healthy older adults could experience reduced CRF based on our prior observations. In the second paragraph of the “Introduction” section (Line 43-48, Page2-3), we explained that age serves as an independent risk factor for the decline in CRF, not only diseases such as hypertension, diabetes, etc. 2) Healthy older adults with declined CRF may not exhibit obvious symptoms in a relatively quiet state. However, when engaging in essential daily activities such as climbing stairs, walking long distances for shopping, or participating in physical exercise, they may experience limitations, inevitably impacting their quality of life and independence. 3) Consequently, when addressing the issue of reduced CRF, healthy older adults are often overlooked compared to groups with underlying diseases. If moxibustion proves effective in enhancing the CRF of healthy older adults, it would bear significant importance in further enhancing their quality of life.

2. Inclusion and exclusion criteria should be provided more clearly.

Our response: We really appreciate your kind comment. It has come to our attention that our inclusion and exclusion criteria were somewhat unclear, with some content being repetitive. 

Following careful deliberation, we have revised and redefined the inclusion and exclusion criteria. We added an inclusion criterion for healthy older adults, and deleted various disease conditions in the exclusion criteria to avoid duplication. 

It is important to highlight that we have added an exclusion criterion according to the recommendation of another reviewer: “Participants with previous moxibustion treatment experience”. Given that real moxibustion treatment induces a noticeable warm sensation, unlike sham moxibustion, participants with prior moxibustion experience may discern the difference, potentially compromising the integrity of the study results. For specifics, please refer to Line 107-121, Page 5-6.

3. Sample size calculation: this trial has 3 arms but sample calculation was done by 2 arms design. Adding additional 35 patients is missing reasonability. 

Our response: Thank you for bringing attention to this crucial issue. There does appear to be an inconsistency in calculating the sample size using this method. We have re-evaluated it using G-power software version 3.1.9.4. Sample size determination was based on one-way analysis of variance (ANOVA) comparing the VO2peak value of three different groups. A Total sample of 107 participants is required to detect a significant difference with an effect size of 0.35 and 90% power at 5% level of significance (see picture below). Considering an estimated dropout rate of 15%, a total of 126 participants are needed, with 42 for each group. We have revised the “Sample size estimation section” (Line 122-130, Page 6) accordingly. Simultaneously, all other mentions of sample size in the manuscript have been adjusted, including the study flow chart (Fig 2).

Reviewer #2: 

This is a design paper for a study examining the improvement of exercise tolerance through moxibustion.

Our response: Thank you for your careful review of this article. Your questions and comments are of great value to the improvement of our study protocol.

The reviewer has some questions for this study.

1. Aren't people with experience in moxibustion excluded? If subject with experience in moxibustion is assigned to the Sham group, is there a possibility that the subject will realize that the sham procedure is being used?

Our response: We really appreciate this great question! Given that moxibustion treatment induces a noticeable warm sensation, unlike sham moxibustion, subjects with prior moxibustion experience may discern the difference, potentially compromising the integrity of the study results. Therefore, as a precautionary measure, we propose adding an exclusion criterion “With previous moxibustion history or experience” to exclude participants with previous moxibustion experience. The pertinent modification is made in Line 115, Page 6 of the manuscript.

2. Isn't the effect of moxibustion different depending on the practitioner? Are differences between practitioners taken into consideration? 

Our response: Thank you for your insightful question. Although moxibustion is less operator-dependent compared to acupuncture, we recognize that variations may arise due to different operators. As far as we know, the factors influencing moxibustion's effectiveness primarily encompass different operators, type of moxibustion, chosen acupoints, moxibustion duration, frequency, and etc. In order to control the influence of these factors on the trial results, 

first, we will minimize the number of practitioners and provide training to standardize their techniques, and homogenize the moxibustion operation process as much as possible, thereby mitigating the potential impact. We have an explanation regarding this under the “Data collection, monitoring, and quality control” section, please check Line 326-330, Page 16 for details.

Second, we meticulously control the other influencing factors, ensuring consistency across each group, including the utilization of the same moxa stick manufacturer, same acupoints, unified session duration and frequency, etc. 

3. What kind of consideration is given to placing moxibustion at the exact position of the acupuncture points?

Our response: Thanks for your question. We believe this process involves two steps: accurately locating the acupuncture points and precisely positioning the moxibustion device on the acupoints. 

For first step, standardized and precise methods for positioning acupoints are available. We will refer to the 2006 National Standard of the People's Republic of China, "Acupoint Names and Locations" (GB/T12346-2006). The acupoint positioning details were elucidated in the “Moxibustion group” section (Line 163-169, Page 8) of the manuscript. 

For second step, following accurate acupoints positioning, the practitioners use a marker to mark the acupoint locations on the local skin and subsequently secure the moxibustion equipment on the selected acupoints. Simultaneously, we utilize adhesive tape to firmly affix the moxibustion equipment, preventing any displacement during the moxibustion process.

In response to your query, we have made pertinent adjustments to the description of the moxibustion operation method to make it clearer. Please check Line 173-193, Page 9 for specific details.

4. What are the possible reasons for withdrawal from research? I think there is a possibility that moxibustion cannot be continued due to pain etc. 

Our response: We appreciate your insightful question. In clinical studies, it is essential to establish criteria for both dropout and withdrawal. For example, if a participant ceases treatment for personal reasons, it is categorized as dropout. Conversely, if the investigators request the subject's withdrawal from the study due to severe moxibustion-related side effects, it is classified as withdrawal.

We acknowledge that our previous explanations were not sufficiently clear. We only used "Dropout" as the heading, while the content also encompasses instances of withdrawal, such as severe side effects, inadequate treatment compliance, significant alterations in lifestyle habits, and weight changes exceeding 5 kilograms, etc. (Line 224-231, Page 11). Consequently, we have revised the heading to “Dropout and withdrawal", and listed different situations in different categories. Please refer to Line 217-231, Page11 for specific details.

As for the participants you mentioned who withdraw from the study due to pain, we believe it is less likely as our moxibustion protocol is designed to impart a sensation of warmth to the subjects, not pain. Regardless, we appreciate your questions, as they led us to identify an issue in this section and correct the content.

---

## [Decision Letter · Decision Letter 1]

20 Mar 2024

Moxibusiton for declined cardiorespiratory fitness of apparently healthy older adults: a study protocol for a randomized controlled trial

PONE-D-23-40936R1

Dear Dr. Sun,

We’re pleased to inform you that your manuscript has been judged scientifically suitable for publication and will be formally accepted for publication once it meets all outstanding technical requirements.

Kind regards,

Yoshihiro Fukumoto

Academic Editor

PLOS ONE

Additional Editor Comments (optional):

Reviewers' comments:

Reviewer's Responses to Questions

**Comments to the Author**

1. Does the manuscript provide a valid rationale for the proposed study, with clearly identified and justified research questions?

Reviewer #1: Yes

Reviewer #2: Yes

2. Is the protocol technically sound and planned in a manner that will lead to a meaningful outcome and allow testing the stated hypotheses?

Reviewer #1: Yes

Reviewer #2: Yes

3. Is the methodology feasible and described in sufficient detail to allow the work to be replicable?

Reviewer #1: Yes

Reviewer #2: Yes

4. Have the authors described where all data underlying the findings will be made available when the study is complete?

Reviewer #1: Yes

Reviewer #2: Yes

5. Is the manuscript presented in an intelligible fashion and written in standard English?

Reviewer #1: Yes

Reviewer #2: Yes

6. Review Comments to the Author

You may also provide optional suggestions and comments to authors that they might find helpful in planning their study.

Reviewer #1: The authors addresed my comments about rationale, inclusion/exclusion criteria, and sample sizea dequately. I do not have further comments at this point.

Reviewer #2: The revised manuscript has been improved based on my comments. I have no more comments for the revised manuscript.

7. PLOS authors have the option to publish the peer review history of their article (what does this mean?). If published, this will include your full peer review and any attached files.

Reviewer #1: No

Reviewer #2: No

---

## [Editor Report · Acceptance letter]

27 Mar 2024

PONE-D-23-40936R1 

PLOS ONE

Dear Dr. Zhang, 

I'm pleased to inform you that your manuscript has been deemed suitable for publication in PLOS ONE. Congratulations! Your manuscript is now being handed over to our production team.

Kind regards, 

on behalf of

Dr. Yoshihiro Fukumoto 

Academic Editor

PLOS ONE